# Learning to See Physical Properties with Active Sensing Motor Policies

**Gabriel B. Margolis**    **Xiang Fu**    **Yandong Ji**    **Pulkit Agrawal**
Improbable AI Lab, Massachusetts Institute of Technology
https://gmargo11.github.io/active-sensing-loco

**Abstract:** To plan efficient robot locomotion, we must use the information about a terrain's physics that can be inferred from color images. To this end, we train a visual perception module that predicts terrain properties using labels from a small amount of real-world proprioceptive locomotion. To ensure label precision, we introduce *Active Sensing Motor Policies* (ASMP). These policies are trained to prefer motor skills that facilitate accurately estimating the environment's physics, like swiping a foot to observe friction. The estimated labels supervise a vision model that infers physical properties directly from color images and can be reused for different tasks. Leveraging a pretrained vision backbone, we demonstrate robust generalization in image space, enabling path planning from overhead imagery despite using only ground camera images for training.

## 1 Introduction

In recent years, legged locomotion controllers have exhibited remarkable stability and control across a wide range of terrains such as pavement, grass, sand, ice, slopes, and stairs [1, 2, 3, 4, 5, 6, 7, 8]. State-of-the-art approaches use sim-to-real learning with a combination of proprioception and depth sensing to perceive the ground beneath the robot or obstacles around it [5, 7, 8, 9, 10, 11, 12, 13, 14] but have discarded a valuable signal about terrain's physical properties that is conveyed by color images. To utilize this information, some works learn to predict locomotion performance or interaction properties from terrain appearance using data collected in the real world [15, 16, 17, 18]. However, the learned representations in these works are task- or policy-specific. Instead, we propose directly predicting the terrain's physical properties (e.g. friction, roughness) that (a) can be simulated and (b) are invariant to the policy and task. With this approach, we can learn a cost map from simulated rollouts to inform traversal planning when performing a new task (like payload dragging) or optimizing a new objective (like a preference for speed vs energy efficiency).

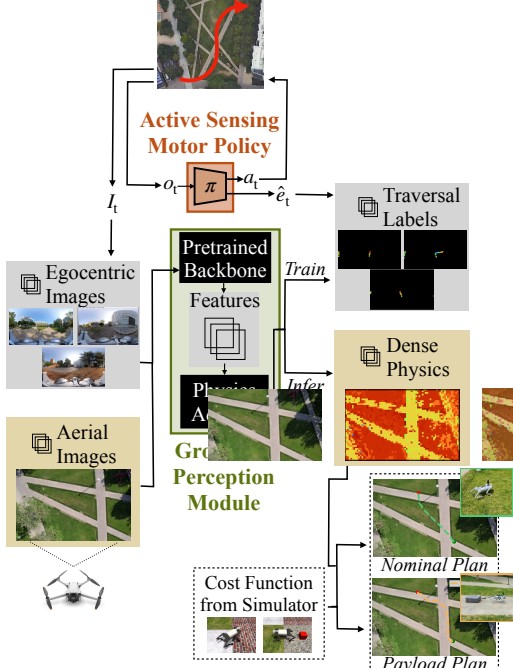

Figure 1: **Learning to see how terrains feel.** We propose (1) learning an optimized gait for collecting informative proprioceptive terrain labels that are (2) used to supervise training for a vision module, which can (3) be used for navigation planning with new tasks and image sources.

A natural way of estimating the terrain's physical parameters during data collection is by training a neural network to predict them from the proprioceptive sensor history, supervised by the ground truth labels available in simulation [4, 5]. We discovered that the estimates obtained through this

7th Conference on Robot Learning (CoRL 2023), Atlanta, USA.

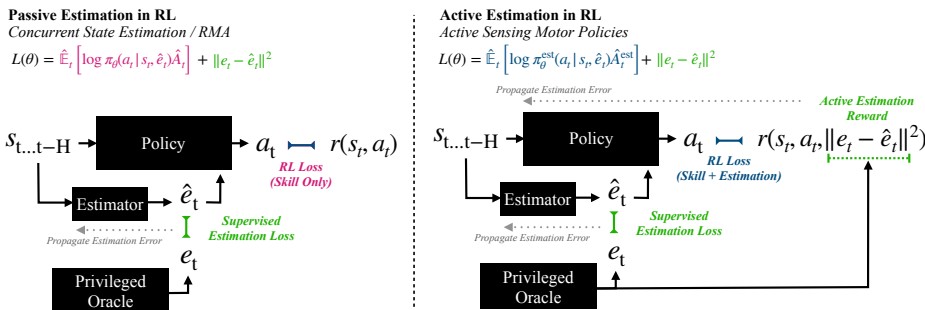

Figure 2: **Active Sensing Motor Policies optimize for estimation.** Unlike passive methods (left), which estimate the state only to the extent that it is observable as a byproduct of control relevance, Active Sensing Motor Policies (right) directly incentivize improved state estimation through the advantage function. This incentivizes a policy to adopt information-gathering behaviors, like intentionally swiping the robot's foot during legged locomotion, to improve the estimate quality.

approach can be imprecise because the locomotion behavior often makes the terrain properties hard to predict. Therefore, unlike prior works in terrain perception that predict the terrain character or traversal cost from passive data [15, 16, 17, 18], we propose training a specialized data collection policy that directly optimizes for terrain property estimation. This *Active Sensing Motor Policy* (ASMP) learns emergent locomotion behavior, such as dragging the feet on the ground to better estimate friction, and improves the informativeness of its proprioceptive traversals.

We use the improved data obtained through ASMP as self-supervision to learn a visual perception module that predicts terrain material properties. The same model can inform efficient plans for nominal locomotion and for dragging objects by considering the impact of terrain properties on traversal cost. Because the robot is low to the ground, its onboard cameras only provide enough range for local planning. Sometimes the robot is in a position where it must plan its motion using only the information in front of it, but in other cases, it might have access to some global information about the environment. Therefore, we also consider the scenario of a teamed drone that flies above the legged robot and provides an extended view of the environment. Despite being trained solely with images captured from an onboard camera, our resulting model can also be evaluated to predict terrain properties using images from various cameras and viewpoints, which allows this type of global planning to succeed.

## 2 Method

Our approach consists of the following stages, which are also illustrated graphically in Figure 1:

1. **Active Sensing**: We estimate the terrain dynamics parameter, $e_t$, from the proprioceptive sensor history during an initial blind traversal. Our *Active Sensing Motor Policy* (ASMP) crucially provides better-calibrated estimates than the baseline policy. In our experiments, the estimated parameter $e_t$ is the ground friction coefficient, the ground roughness magnitude, or both. (Section 2.1)

2. **Self-Supervised Vision Learning**: Using labels of $e_t$ recorded from the real-world traversal of the robot, we learn a function, $\hat{e} = f(\mathbf{I})$, that predicts the per-pixel value of $e_t$ for a given image $\mathbf{I}$. The labels for training are only available at the pixels corresponding to the places the robot traversed, but the resulting model can be queried to predict the terrain parameter at any pixel. (Section 2.2)

3. **Cost Function Learning**: To inform planning, we learn cost functions that relate the terrain dynamics to various performance metrics. First, we create terrains with a range of $e_t$ values in simulation. Then, we perform rollouts in simulation to measure a cost function $C(e_k)$ that relates dynamics parameters to performance. We learn a separate cost function for each task. (Section 2.3)

4. **Dynamics-Aware Path Planning**: Combining (2-3), we compute cost maps directly from color images and use them for path planning. (Section 2.4)

## 2.1 Active Sensing Motor Policies: Learning Whole-Body Active Estimation

In learning control policies under partial observations, it is commonplace to train with an implicit [1, 2, 3, 5, 8] or explicit [4, 19] incentive to form representations within the policy network that correspond to the unobserved dynamics parameters. Consider the concurrent state estimation framework of Ji et al. [4], under which a state estimation network is trained simultaneously with the policy network to predict the unobserved parameters. The predictions of the state estimation network are concatenated with the rest of the observation to construct the policy network input. This approach optimizes a two-part loss consisting of the standard policy objective and the state estimation error:

$$L(\theta, \theta') = \hat{\mathbb{E}}_t \left[ \log \pi_\theta(a_t | s_t, \hat{e}_t) \hat{A}_t \right] + \| e_t - \hat{e}_{\theta'}(s_t) \|^2.$$

This has been empirically shown to yield better policy performance in environments with randomized dynamics or unobserved state variables [4].

In the formulation above, the estimation error is used to update the state estimator weights $\theta'$, but not the policy weights $\theta$. This does not incentivize the *policy* to adjust its actions to improve estimation performance beyond what is required for control. Typically, this is no problem because it allows the policy to maximize its performance at the current control task. However, our end goal is to use the output of the state estimator to train a visual perception module that may be reused with other controllers and tasks. To support this, the labels should be as accurate as possible even when that is not necessary for control. To obtain the most accurate perception module, we would like a mechanism to improve the state estimate quality of the proprioceptive data collection policy as much as possible by adapting the policy's behavior. To this end, we propose *Active Sensing Motor Policies* in which the policy $\pi^{\text{est}}$ is trained with an additional *estimation reward*: $r_{\text{est}} = c \cdot \exp \left( \| e - \hat{e} \|^2 \right)$. Figure 2 illustrates the policy architecture.

In practice, we observe that an Active Sensing Motor Policy that is rewarded for estimating the ground friction coefficient slides one foot along the ground or swipes it vigorously to improve the friction coefficient observability in the state history.

## 2.2 Grounding Visual Features in Physics from Real-world Experience

We collect paired proprioceptive and vision data from the state estimation policy in the real world in order to learn about the relationship between visual appearance and terrain physics. Specifically, we collect data of the form $(\mathbf{I}, \hat{e}, \mathbf{x})_t$ where $\mathbf{I}$ is a camera image, $\hat{e}$ are the estimated dynamics parameters and $\mathbf{x}$ is the position and orientation of the robot in a fixed reference frame. We obtain $\mathbf{x}$ by training an additional 2D output of the final MLP layer in our learned state estimator to predict the displacement in the ground plane of the base from its location at the previous timestep, $\Delta\mathbf{x}$, and then integrate the estimated displacements. The integrated estimates $\mathbf{x}$ will drift over time, but we will only rely on them over a short time window. This alleviates the need for a separate odometry algorithm to estimate the robot's state.

Using the camera intrinsic and extrinsic transform, we project the relative positions of the robot in the past and future into each camera image frame. We restrict the positions to those between $1\,\mathrm{m}$ and $5\,\mathrm{m}$ from the robot along the traversal path so that they are neither too far away to see nor so close as to be obstructed from view by the robot's own body. We label each of the projected robot positions with the estimated dynamics parameters $\hat{e}$ that the robot felt when it walked there. This yields a corresponding label image $\mathbf{I}_t^e$ for each color image $\mathbf{I}$ where the traversed pixels are labeled with their measured dynamics.

For each color frame $\mathbf{I}_t$, we use the pretrained convolutional backbone [20] to compute a dense feature map. Similar to the procedure that Oquab et al. [21] used for depth estimation, we discretize the labels $\hat{e}_t$ into 20 bins and train a single linear layer with cross-entropy loss where the inputs are the features of one patch and the outputs are the logits of the patch's $\hat{e}_t$ label from proprioception.

## 2.3 Cost Function Learning: Connecting Physics Parameters to Affordances

The impact of terrain properties on robot performance is task-dependent: for example, a robot dragging an object may face distinct constraints that inhibit its traversal on some terrains, compared to a robot without any payload. To use our vision module for planning, we must establish a mapping between terrain properties and robot performance for each task. We propose a simple procedure for extracting a task cost function from simulated data to demonstrate that our perception module can be useful in planning for multiple tasks, which we refer to as "operating modes". We sample simulated terrains with a variety of terrain properties $e_t$ and command a locomotion policy from prior work [19] to walk forward at $1\,\mathrm{m/s}$. We record the actual resulting velocity achieved on each terrain. We evaluate the mean realized velocity

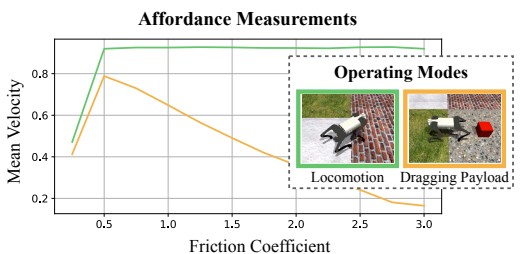

Figure 3: **Locomotion affordances.** We measure the dependence of locomotion performance $(1\,\mathrm{m/s})$ on terrain friction in two different operating modes. In free locomotion, the controller maintains the target velocity across a range of friction coefficients, except for the lowest friction. In contrast, when dragging a weighted box, the robot slows down as the terrain friction increases.

for multiple operating modes: (1) locomotion, (2) payload dragging. We construct a cost function for each operating mode as the average time spent traversing one meter of a given terrain. Minimizing this cost function during path planning will yield an estimated shortest-time path. While we focus on time-optimal payload dragging as an example, (1, 2) could be any combination of task and metric as long as their relation to terrain properties can be evaluated in simulation.

## 2.4 Integrated Dynamics-Aware Path Planning from Vision

Our perception module (Section 2.2) runs in real-time $(2\,\mathrm{Hz})$ using onboard compute. Although it was trained using images from a 360-degree camera, the resulting pixel-wise friction estimator can be evaluated in images from other cameras including the robot's onboard fisheye camera and an overhead drone. This is useful because the perception module can remain useful when deployed on a new robot or evaluated from a new viewpoint.

One possible scenario for carrying ground objects across a long distance is that of a drone-quadruped team. In these cases, we can directly evaluate our grounded vision module in overhead images to obtain a pixel-wise friction mask. Then, considering the robot's operating mode, we compute the cost associated with each pixel using the corresponding cost function determined from simulation (Section 2.3). Given this overhead cost map, we use the A$^*$ search algorithm [22] to compute the minimum cost traversal path for the current operating state.

## 2.5 System Setup

**Robot**: We use the Unitree Go1 robot, a 12-motor quadruped robot stands $40\,\mathrm{cm}$ tall. It has an NVIDIA Jetson Xavier NX processor, which runs the control policy and the vision module. For payload dragging experiments, the robot's body is connected to an empty suitcase using a rope.

**360 Camera**: We use a Insta360 X3 360 action camera mounted on the robot to collect images for training the perception module. This camera provides a $360°$ field of view. Before the image data is used for training, we use the Insta360 app to perform image stabilization, which takes about two minutes for data collected from a ten-minute run.

**Training Compute**: We perform policy training, video postprocessing, and vision model training on a desktop computer equipped with an NVIDIA RTX 2080 GPU.

**Drone Camera**: For planning from overhead images, we record terrain videos using a DJI Mini 3, a consumer camera drone.

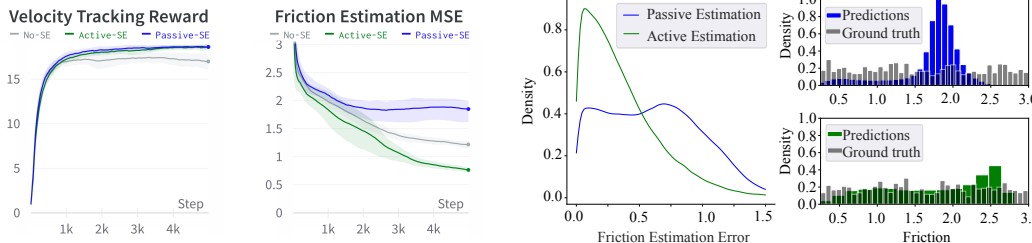

(a) Performance and estimate quality during training.  (b) Distribution of friction estimates at convergence.

Figure 4: **Learning active estimation.** Active Sensing Motor Policies (`Active-SE`) automatically learn motor skills (e.g. dragging the feet) that improve observability of the environment properties.

## 3 Results

### 3.1 Interaction among Estimation, Adaptation, and Performance

*Observing supervised internal state estimates improves proprioceptive locomotion.* Affirming the results of Ji et al. [4], we train a state estimation network using supervised learning to predict privileged information (the ground friction coefficient and terrain roughness parameter) from the history of sensory observations. When the policy is allowed to observe the output of this state estimation network (`Passive-SE`), the policy training is more stable and results in a more performant final policy than when the state estimate is not observed (`No-SE`) (Figure 4).

*Observing passive state estimates can degrade the state observability.* We analyze the error distribution of the learned state estimator in `Passive-SE` and `No-SE` policies (Figure 4). It may be surprising that the friction estimation error of the more-performant `Passive-SE` policy is *worse* than that of the less-performant `No-SE` policy. We suggest a mechanistic explanation for this behavior: Supposing some irreducible sensor noise, two terrains of different frictions will only be distinguishable if they make the robot slip in sufficiently different ways. However, a control policy with a better adaptive facility is more likely to avoid slipping across a wide range of ground frictions. Therefore, in the more adaptive policy, slip will occur less intensely, and as a result, the observability of the ground friction coefficient will degrade.

*Our method, ASMP, produces the best privileged state observability.* We train an active sensing motor policy (`Active-SE`) to intentionally measure the friction as described in Section 2.1. (The full reward function for each policy we trained is provided in the appendix.) We find that the `Active-SE` policy provides the most accurate friction estimates among the three architectures (Figure 4). Therefore, as we will further show, it is the superior policy for supervising a task-agnostic physical grounding for vision.

### 3.2 Learning to See Friction

**Evaluation in Simulated Environment.** We collect five minutes of simulated data on four terrains: ice, gravel, brick, and grass, assigning them arbitrary friction coefficients of $\mu = \{0.25, 1.17, 2.08, 3.0\}$ respectively. Figure 10 compares the resulting visual perception module learned from the policies performing passive vs. active estimation. Qualitatively, the vision module learned from passive data learns to see ice but fails to distinguish between higher-friction terrains (gravel, brick, and grass). This makes sense as Figure 3 shows that frictions in this range have less influence on the performance of locomotion. In contrast, the vision module trained on data from our Active Sensing Motor Policy correctly learns to distinguish all four terrains. Quantitatively, ASMP results in lower dense prediction loss on images from a held-out test trajectory (Figure 10, Appendix).

**Real-world Training.** We collect 15 minutes of real-world traversal data spanning diverse terrains: grass, gravel, dirt, and pavement. Following the procedure in Section 2.2, we project the traversed

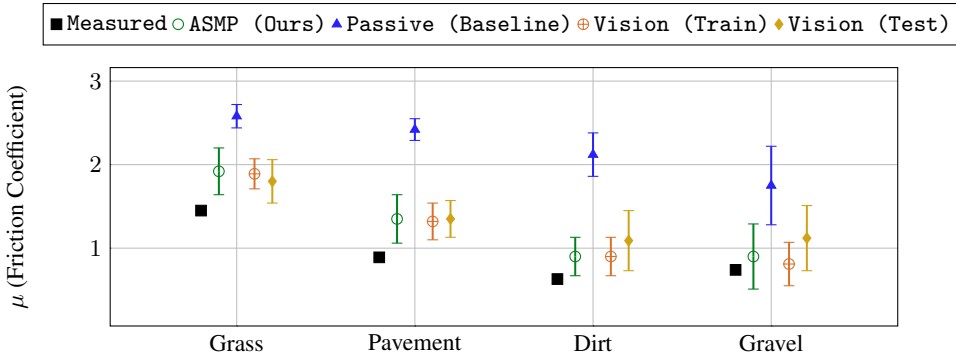

Figure 5: **Real-world friction sensing performance with proprioception and vision.** `Measured` values are directly measured by a dynamometer. The predictions from our proposed `ASMP (Ours)` agree more strongly with the dynamometer measurements than the baseline `Passive (Baseline)`. `Vision (Train)` shows the generalization of visual prediction to un-traversed patches in the training images from the onboard camera; (`Vision (Test)`) shows the generalization to unseen patches and viewpoints by evaluating on drone footage. We use manual segmentation maps (Appendix Figure 9) to match pixel predictions to terrains. Error bars indicate one standard deviation.

points into the corresponding camera images and train a linear head on top of a convolutional backbone trained for segmentation [20] to predict the terrain friction estimate for each traversed patch.

To evaluate estimation performance in the real world, we manually label image segments in a subset of train and test images containing grass, pavement, dirt, or gravel and compute the distribution of proprioceptive and visual friction predictions for each (Figure 5). To obtain a ground truth friction value for comparison, we use dynamometer to measure the weight of a payload made of robot foot material and its drag force across each terrain. The proprioceptive estimates from ASMP are much closer to the dynamometer measurements than the estimates from the passive baseline. They do not match perfectly, suggesting a small but measurable sim-to-real gap in the robot dynamics or terrain modeling. They agree with the dynamometer measurements on the ordering of terrains from most to least slippery. The grounded vision module is close to the distribution of proprioceptive estimates for both train and test images, with increased variance in test images.

### 3.3 Integrated Planning

**Cost Function Evaluation.** We define a cost metric for the locomotion policy from [19] as the distance traveled per second when commanded with a speed of $1.0 \, \mathrm{m/s}$. We evaluate this metric in simulation by averaging the performance of 50 agents simulated in parallel for $20 \, \mathrm{s}$ on terrains of different friction coefficients ranging from a lower limit of $\mu = 0.25$ to an upper limit of $\mu = 3.0$. This procedure is performed once with the robot in nominal locomotion and again with the robot dragging a $1.0 \, \mathrm{kg}$ payload. Figure 3 shows the measured result; both tasks yield poor performance on extremely slippery terrain, but on higher terrains, the robot dragging a payload slows down while the free-moving robot adapts to maintain velocity. Knowledge of the ground's physical properties motivates a difference in high-level navigation decisions between the two tasks.

**Path Planning and Execution.** We plan paths for locomotion and payload dragging and execute them via teleoperation to evaluate whether the predicted preferences hold true in the real world. We fly a drone over the same environment where the vision model was trained and choose a bird's-eye-view image that includes grass and pavement. We estimate the friction of each pixel and from this we compute the associated cost for locomotion and payload dragging. Then we use A* search to compute optimal paths. The optimized paths and traversal result are shown in Figure 6. In agreement with the planning result, it is preferred to remain on the sidewalk while dragging the payload and cut directly across the grass when in free locomotion.

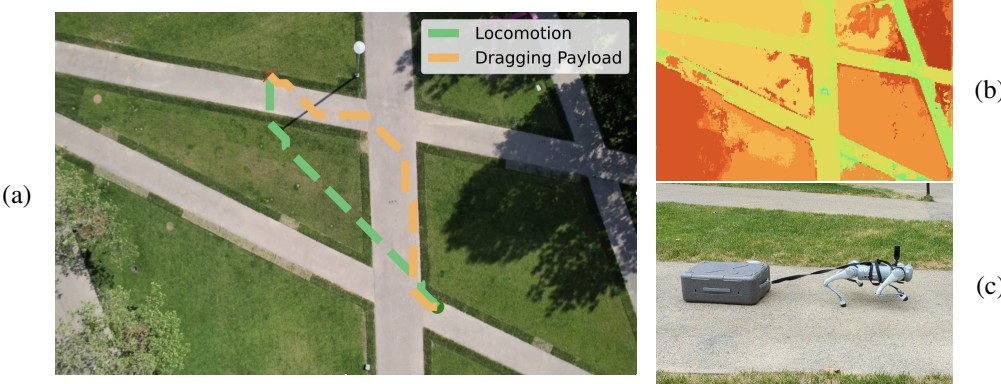

| Operating Mode | Metric | Cross Grass | Stay on Sidewalk |
|---|---|---|---|
| Dragging Payload | Time (s) | $48 \pm 1$ | $45 \pm 1$ |
| Locomotion | Time (s) | $23 \pm 1$ | $26 \pm 0$ |

Figure 6: **Path planning in overhead images.** (a) We use the learned vision module to plan navigation in overhead images of terrain. (b) The vision module is only trained using first-person views from the robot but can infer the terrain friction with a different camera model and viewing pose. (c) We teleoperate the robot across both planned paths in both locomotion modes. The preference among paths in the real world matches the planning result from our pipeline.

## 4 Related Work

Self-supervised traversability estimation has been studied previously for the navigation of wheeled and legged robots. Some works have focused on the direct estimation of a traversability metric, a scalar value quantifying the cost of traversing a particular terrain [16, 17, 23]. These approaches are specialized to the robot's traversal capability at the time of data collection, implying that a change in the policy or task may necessitate repeated data collection to train a new vision module.

Other works have demonstrated self-supervised terrain segmentation from proprioceptive data [15, 24, 25]. Wu et al. [24] demonstrated that proprioceptive data from a C-shaped leg equipped with tactile sensors may be sufficient to classify different terrains. Wellhausen et al. [15] took supervision from the dominant features of a six-axis force-torque foot sensor during traversal and trained a model to densely predict a "ground reaction score" from color images to be used for planning. Łysakowski et al. [25] also demonstrated that terrain classification from proprioceptive readings could be performed in an unsupervised manner on a full-scale quadruped and showed that this information could be used as an additional signal to improve localization. Our work differs from these in that (1) we do not use any dedicated sensor in the foot but predict the terrain properties using only standard sensors of the robot's ego-motion, and (2) thanks to our Active Sensing Motor Policies, we can directly predict the terrain properties instead of a proxy score, which allows us to compute the cost function in simulation for multiple scenarios as in Section 2.3.

Another possibility is directly predicting which locomotion skill to execute from visual information[18, 26]. Loquercio et al. [26] learned to predict the future latent state of the policy [2] from a front-facing camera image to improve low-level control performance in stair climbing. An advantage of their approach is that it does not require the choice of an explicit terrain parameterization, but this comes at the cost that its visual representation is specialized to the latent of a single motor policy, so it cannot be reused for new policies or operating states, and predicting the next latent is only meaningful for egocentric images, so it cannot be used for novel viewpoints, as in drone-quadruped teaming or planning from satellite imagery. Yang et al. [18] trained a semantic visual perception module for legged quadrupeds using human demonstrations.

The resulting system imitated an operator's response to different terrains, controlling velocity and gait. This relies on a human operator to predict the terrain properties during the demonstration. Other work has learned general navigation through supervised learning on diverse robotic platforms, including legged robots [27, 28, 29]. These works train an omni-policy for all robots and environments, enabling interesting zero-shot generalization but not explicitly adjusting for embodiment, operating state, or camera viewpoint variation.

Several works on wheeled robots visually estimate the geometry or contact properties of the terrain through self-supervision or hand-designed criteria and then compute the traversal

| Estimation Mode | Friction Loss | Roughness Loss | Torque Penalty |
|---|---|---|---|
| Passive | 1.00 | 1.00 | −0.34 |
| Friction | 0.47 | 1.06 | −0.87 |
| Roughness | 0.99 | 0.72 | −0.84 |
| Joint Fr.+Ro. | 0.49 | 0.80 | −1.18 |

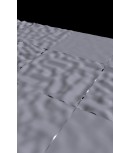

Figure 7: **ASMP for multiple physical parameters.** Friction and roughness estimates are improved by ASMP, even when both parameters are jointly targeted. We report estimation loss for passive estimation (`None (Passive)`), active estimation of each parameter separately (`Friction`, `Roughness`), and active estimation for both parameters in a single policy (`Joint Fr.+Ro.`). Variation in torque reflects that a change in motor strategy enabled the improved estimation.

cost from these metrics [30, 31, 32, 33, 34, 35, 36, 37, 38]. Compared to legged robots, wheeled robots have a limited variety of traversal strategies. Consequently, the question of selecting a locomotion controller to gather the most informative self-supervision data has not been directly addressed. Active perception, in which a robot agent acts to increase environmental observability, suggests a solution. This approach has been applied to vision systems [39, 40, 41] , and more recently has been extended to include physical interaction [42, 43, 44, 45]. This inspired our approach to the controller selection issue in labeling vision with proprioception for legged robots.

## 5    Discussion and Limitations

Our work assumes a mapping between the estimated terrain properties and the robot's performance. Friction affects the slip of the robot's feet against the ground and the drag force of payloads and other objects, so it is an interesting source of performance variation for practical locomotion tasks. To account for other parameters besides friction that vary in the environment, our framework can potentially be extended to include them. For example, Figure 7 shows that ASMP successfully enhances the estimation of a terrain roughness parameter in addition to friction.

In general, ASMP may be applied under two conditions: (1) a history of proprioceptive readings is sufficient to infer the parameter of interest, and (2) the parameter of interest can be effectively simulated. If these conditions are not met, a different technique besides ASMP may be necessary to collect training data. Additionally, to train our vision module, we assume terrains with different properties are visually different. If some parameters do not have an impact on the terrain's visual appearance, it may be impossible to learn a vision module of the form we propose for those parameters. If the mapping varies quickly over time, future work could explore representing uncertainty or performing fast online adaptation of the estimates to the current environment based on new proprioceptive information. Finally, our pipeline does not yet account for terrain geometry and occlusions. In the future, ASMP may also be useful to complement data from other sensors like LiDAR to achieve the most detailed perceptual representation that includes geometric information.

## 6    Conclusion

Proprioceptive self-supervision is a promising data source for robots to learn about the relationship between vision and physics. In this work, we exposed that the quality of proprioceptive supervision can be strongly influenced by the style of the motor policy acquired through reinforcement learning. We proposed a novel technique, Active Sensing Motor Policies, and show that it improves the proprioceptive estimation quality and the corresponding performance of a grounded vision module that is reusable for new sensor configurations and physical tasks.

**Acknowledgments**

We thank the members of the Improbable AI lab for the helpful discussions and feedback on the paper. We are grateful to MIT Supercloud and the Lincoln Laboratory Supercomputing Center for providing HPC resources. This research was supported by the DARPA Machine Common Sense Program, the MIT-IBM Watson AI Lab, and the National Science Foundation under Cooperative Agreement PHY-2019786 (The NSF AI Institute for Artificial Intelligence and Fundamental Interactions, http://iaifi.org/). We acknowledge support from ONR MURI under grant number N00014-22-1-2740. This research was also sponsored by the United States Air Force Research Laboratory and the United States Air Force Artificial Intelligence Accelerator and was accomplished under Cooperative Agreement Number FA8750-19-2-1000. The views and conclusions contained in this document are those of the authors and should not be interpreted as representing the official policies, either expressed or implied, of the United States Air Force or the U.S. Government. The U.S. Government is authorized to reproduce and distribute reprints for Government purposes, notwithstanding any copyright notation herein.

**Author Contributions**

- **Gabriel B. Margolis** ideated, implemented, and evaluated Active Sensing Motor Policies and shared ideation and implementation of the vision module and overall experimental design.

- **Xiang Fu** shared ideation and implementation of vision module and overall experimental design.

- **Yandong Ji** contributed ideas and supported infrastructure development during the project.

- **Pulkit Agrawal** advised the project and contributed to its conceptual development, experimental design, positioning, and writing.

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
