# OpenReview forum: "Learning to See Physical Properties with Active Sensing Motor Policies"
_robot-learning.org/CoRL/2023/Conference — CoRL 2023 Poster_

### Official Review · Reviewer_knX2 · 2023-07-05

**Confidence:** 4
**Originality:** Good
**Technical Quality:** Very Good
**Clarity Of Presentation:** Very Good
**Impact:** 4

**Recommendation:**

Weak Accept: I recommend accepting the paper, but will not argue for my recommendation if the majority of other reviewers have a different opinion.

**Review:**

The rebuttal period has very much strengthened the paper and the authors have adequately responded to the open questions.

STRENGTHS:
The task of environmental parameter estimation is generally a very important one to work on, and by explicitly estimating the friction, the approach presented here is more versatile than, for example, RMA, which has an implicit representation in a latent space and does not optimize for observability. The approach shown here is somewhat similar to drivers trying out the traction on snow or ice at a slow speed to calibrate their perception of the road and adjust accordingly. Therefore, the reviewer believes that the task is highly relevant to robotics and the approach presented is potentially impactful.

The paper clearly demonstrates that the presented ASMP approach enhances the estimation of the ground friction coefficient. The real-world experiments comparing the friction estimates from an active and a passive policy with the measured ground-truth value prove the effectiveness of ASMP to improve the observability of the given ground parameter. Additionally, the good agreement between ground-truth friction coefficient (obtained through a dynamometer) and the ASMP estimate highlight the accuracy of the overall method. The real-world experiments also highlight that the vision system trained on the ASMP observations is outputting coherent friction values based on the camera images.

The authors hint at the generalizabilty of the method to other parameters beyond friction estimation and show simulation results of terrain roughness estimation. This is a key point as ASMP can, in theory, be applied to almost any mobile robot that needs to estimate a certain parameter of its environment. The reviewer therefore believes that the method potentially has significant impact.

WEAKNESSES:
The experimental validation of the friction coefficient estimation is convincing, however the demonstrated application in pulling a crate across various terrains is not. First, as pointed out by other reviewers, the object that is being pulled by the robot also interacts also with the ground and hence is subject to some friction as well. In the results shown, only the robot foot friction is estimated - and there is no neccessity that the object being pulled by the robot behaves similar. Also, unless the robot is severly underpowered for the task, a change in friction should not affect the pulling performance too much as less friction for the legs also likely (but not neccessarily!) corresponds to less friction for the pulled obstacle.

For very similar reasons, it is unclear (perhaps even undefined) how friction of the robot foot and traversal time for a given course are related. This reviewer believes that the parameter estimation performance of the method is its key feature, and less the application to path planning shown in the paper.

**Quality Of The Limitations Section:**

Limitations are addressed clearly

**Questions For Rebuttal:**

- Why is a separate policy better than having a locomotion policy that continuously estimates the friction values (explain the statement in line 29)?
- Using published friction values (or measuring them accurately), how close is the estimate to the ground-truth value in the different real-world scenarios?
- Is there any measure of how consistent and homogenous the estimated friction coefficients are in areas of similar materials?  This is especially interesting in the context of planning as any (false) variations in the friction coefficient might lead to unnecessarily complex paths.
- How is friction related to traversal time?

**Robotics Focus:**

Highly relevant to robotics but no hardware experiments

**Summary Of Paper:**

The work "Learning Physically Grounded Robot Vision with Active Sensing Motor Policies" proposes an approach to train quadrupedal locomotion policies optimized for environment parameter estimation. They demonstrate this by training a controller which optimizes for foot friction coefficient estimation and learns to drag its foot to improve the observability of the friction. The friction estimates are then used to train a network that directly predicts the friction coefficient from images.

**Summary Of Recommendation:**

The direction of the research and the methodology are very interesting and relevant. The presented ASMP method can be used for accurate parameter estimation and the authors demonstrate this with friction coefficient estimation. Furthermore, they hint towards the generalizability by showing (simulated) results for terrain roughness estimation. Overall the method and the experiments are convincing.
The application of the method, path planning when dragging a payload, is less convincing though. Nevertheless this reviewer recommends accepting the manuscript.

---

### Official Review · Reviewer_zcuB · 2023-07-19

**Confidence:** 4
**Originality:** Good
**Technical Quality:** Good
**Clarity Of Presentation:** Very Good
**Impact:** 3

**Recommendation:**

Weak Accept: I recommend accepting the paper, but will not argue for my recommendation if the majority of other reviewers have a different opinion.

**Review:**

Strengths:

- Using RL to improve the estimation of terrain property is interesting and makes intuitive sense. The supplementary video shows an interesting behavior learned by the policy.
- The paper is well-written and easy to follow.
- The figures are illustrative and informative.
- The limitation section is informative.

Weakness:

- It is unclear how effective the proposed method is for estimating other terrain properties.
- It is unclear which one is better: optimizing the terrain properties or optimizing the final policy directly. This seems to boil down to the reward design. The advantage of directly optimizing the final policy is that it can be trained with real-world data, and it considers all environmental factors. On the other hand, the proposed approach has to be trained in simulation, and it is unclear how big the sim2real gap is. It would be more enlightening if the paper elaborates on this and includes comparisons with some recent baselines (e.g., [1, 2, 3, 4, 5]).


**Quality Of The Limitations Section:**

Limitations are addressed clearly

**Questions For Rebuttal:**

- How much sim2real gap is there? Could you elaborate what may hinder the sim2real transfer? It will be more convincing if the paper conducts an experiment that compares the friction parameter estimation in the real world.
- Why friction? Since there will still be error in the friction estimation and there are other factors , the corresponding costmap may not be accurate. Would it be possible to estimate the terrain cost with RL?
- What are the tradeoffs between passive and active sensing? While active sensing could be more accurate, it may also lead to unexpected behaviors or even accidents. Passive sensing is safer since it is more controllable.
- It is unclear if the active sensing approach has significant advantages in the real world since the paper only focuses on friction estimation. A more end-to-end approach may potentially work better because it considers more environmental factors and optimizes the final locomotion policy. Could you elaborate on this?
- How is the learning-based odometry trained? Is it better than the default odometry?

**Robotics Focus:**

Sufficient demonstration on hardware

**Summary Of Paper:**

This paper proposes learning terrain properties via active sensing motor policies (ASMP). ASMP uses a reward term that encourages the policy to produce actions that improve the accuracy of state estimation. In this work, only terrain friction is considered for state estimation. Given the estimated friction parameters, a robot trajectory is projected on to the observations (RGB images), thereby associating each image patch with the corresponding predicted friction parameter. Then, a linear layer is trained to predict the friction parameter from the image patch. The policy is trained in simulation and results show that it produces more accurate friction estimation than the baselines.  For planning, the predicted friction is converted to cost and an off-the-shelf planner computes the optimal path. Finally, the paper conducted a real experiment where a robot would prefer sidewalk over grass if it carries a payload.

**Summary Of Recommendation:**

The paper is well-written and the idea is interesting and makes intuitive sense. I have some questions regarding the advantages of this approach compared to end-to-end policy optimization , its general applicability for other state estimation tasks, and the sim2real gap. I think this is critical to understand the potential of this approach.

**Post rebuttal**
The authors have done a great job of conducting additional experiments and discussing the pros and cons of the proposed approach. I think the findings are valuable to the community. I recommend accepting the paper.

---

### Official Review · Reviewer_dDBh · 2023-07-20

**Confidence:** 4
**Originality:** Very Good
**Technical Quality:** Good
**Clarity Of Presentation:** Very Good
**Impact:** 4

**Recommendation:**

Weak Accept: I recommend accepting the paper, but will not argue for my recommendation if the majority of other reviewers have a different opinion.

**Review:**

The idea of learning a policy to actively explore the environment for better system identification is interesting and the presented policy indeed learns interesting behavior. The entire pipeline to obtain the cost-map seems solid and results show intuitively meaningful behavior on real hardware.

One weakness of the work is that it currently heavily focus on friction coefficient and assumes that a good mapping between friction and affordance exists. However, this may not be true for other terrains such as stairs and it's not clear how well the method can scale up to higher dimensional physical parameters.

In addition, I think one important baseline is missing in the current evaluation, where one directly learns the affordance in real-world, using the same strategy for learning the visually grounded friction model in this work (similar to [1]).

[1]. Where Should I Walk? Predicting Terrain Properties from Images via Self-Supervised Learning.

**Quality Of The Limitations Section:**

Limitations are addressed clearly

**Questions For Rebuttal:**

1) It would be helpful to add some discussions regarding whether a good mapping between friction and affordance exists.
2) Add some analysis/discussion on scalability to higher-dimensional physical parameters identification.
3) Add some comparisons to demonstrate the value of going through friction as an intermediate step vs directly learning the affordance. i.e. use the final policy to collect the same amount of data on the same route used in the current experiment for learning friction, but this time estimate the realized velocity and construct a model to predict per-pixel cost directly from the data. This would also enable the method to handle cases where friction alone cannot predict affordance well.

**Robotics Focus:**

Sufficient demonstration on hardware

**Summary Of Paper:**

The work presents a learning-based system to plan a navigation path for a legged robot in multi-terrain outdoor environments. The core idea is to learn a cost-map of the environment for planning the most efficient route to take. To obtain the cost-map, a three-stage approach is devised: 1) an active sensing motor policy is trained to optimize the system identification error for friction so that the policy takes actions to best estimate the friction of the environment, 2) the active sensing motor policy is then deployed in the real-world to collect a set of image data with sparse friction labels, which is then used to train a model to predict per-pixel friction coefficient, 3) the target locomotion controller is tested in simulation with different friction coefficients to estimate the 'affordance' of the controller at different frictions, which is then used to map the dense friction prediction into an affordance/cost map for the path planning.


**Summary Of Recommendation:**

The paper presents a solid pipeline with interesting ideas and results. However it's missing some key analysis and comparisons. With these resolved I believe it can be a good contribution to the venue.

---

### Official Review · Reviewer_dYv6 · 2023-07-25

**Confidence:** 5
**Originality:** Very Good
**Technical Quality:** Good
**Clarity Of Presentation:** Very Good
**Impact:** 3

**Recommendation:**

Weak Accept: I recommend accepting the paper, but will not argue for my recommendation if the majority of other reviewers have a different opinion.

**Review:**

This paper extends the work of co-trained state-estimator to environment physical properties and generalizes the application of estimated value to task planning. The work also demonstrated that the physical property estimation model can be used for task planning of multiple different tasks. Both active sensing policy training and the observation of physical properties generalize better are novel points proposed in the paper.

The paper is well written and easy to follow with a good amount of description and figures to help illustrate the ideas and procedure.

Though the support of the proposed points can be strengthened (see question for rebuttal section).


**Quality Of The Limitations Section:**

Limitations are addressed clearly

**Questions For Rebuttal:**

1) the analysis of generalization to OOD data (bird-eye view) is not clear: only one example is provided and no intermediate results(such as the vision model output) are shown.

2) the performance of the grounded vision model in terms of estimation accuracy is not discussed. This is important because the ASMP is trained in sim environment and the grounded vision model uses the data generated from it.


**Robotics Focus:**

Sufficient demonstration on hardware

**Summary Of Paper:**

This paper presented a method to train a vision model with physical grounding (image -> friction coefficient of surface) and an application of such a vision model in task specific legged robot navigation path planning.

The work is done in stages: 1) train an active sensing legged robot locomotion policy with auxiliary task of state estimation and environment physical parameter (friction coefficient) estimation; 2) roll out policy in real world and collect images and physical parameter estimation output as training data to train a physically grounded vision model that estimate friction of coefficient from images input; 3) apply the grounded vision model for navigation planning.

Main contribution of the papers are
* Pipeline for training grounded vision models from self-supervision data and apply output to task planning.
* Showed that physical property can be translate to task specific affordance.
* Reward design for co-trained active sensing policy.

**Summary Of Recommendation:**

The paper has clear novel points and the procedure and results are well documented and easy to follow/understand. There are two points that can be addressed to strengthen the claim of the author.

I recommend accepting the paper if the author can address the two questions raised in the rebuttal questions.

---

### Comment · Area_Chair_GDp2 · 2023-08-10
**Discussion period**

Dear authors, the discussion period is until next Tuesday (Aug 15), to make sure there is enough time to engage in a meaningful discussion with the reviewers, I would recommend submitting the rebuttals soon.

-AC

---

> ### Author Response · Authors · 2023-08-12
> **Rebuttal Posted**
>
> Dear AC and Reviewers,
>
> Thank you for your time and patience. We have now posted our responses addressing each review.
>
> We have attached a revised version of the paper to each rebuttal below. Here is a summary of the major updates:
>
> - Added quantitative results evaluating the friction prediction performance in the real world (Fig. 5).
> - Added experiments extending ASMP to estimate an additional terrain roughness parameter (Fig. 7).
> - Revised the introduction and discussion of limitations to cover points raised by reviewers.
> - Uploaded additional media to the project website.
>
> We hope that these changes have addressed the reviewers' concerns and look forward to further discussion in the case of remaining questions.
>
> Best,
> Authors

---

### Decision · Program_Chairs · 2023-08-30

**Decision:**

Accept (Poster)

**Comment:**

The paper introduces active sensing motor policies (ASMP) for learning terrain properties, focusing on terrain friction estimation using actions that enhance state accuracy. After estimating friction from RGB images, a linear layer predicts friction from the image patch. Simulations show ASMP has better friction estimation compared to baselines. For planning, friction estimates are used to calculate costs, which is then used by a standard planner to determine the optimal path. In real-world tests, robots carrying payloads favored sidewalks over grass.

The reviewers unanimously agreed on the quality of this work, including sufficient real-world evaluations. There are some remaining concerns, e.g. reviewers are not sure if the foot friction prediction is helpful for optimizing crate-pulling behavior. Additionally, the authors are encouraged to take in remaining editorial feedback and improve the paper for the camera-ready version.